# Development and Characterization of Thiolated Cyclodextrin-Based Nanoparticles for Topical Delivery of Minoxidil

**DOI:** 10.3390/pharmaceutics15122716

**Published:** 2023-12-01

**Authors:** Ammara Akhtar, Muhammad Khurram Waqas, Arshad Mahmood, Saira Tanvir, Talib Hussain, Mohsin Kazi, Muhammad Ijaz, Mulazim Hussain Asim

**Affiliations:** 1Institute of Pharmaceutical Sciences, UVAS, Lahore 54000, Pakistan; ammarach4567@gmail.com (A.A.);; 2College of Pharmacy, Al Ain University, Abu Dhabi Campus, Abu Dhabi P.O. Box 112612, United Arab Emirates; 3AAU Health and Biomedical Research Center (HBRC), Al Ain University, Abu Dhabi P.O. Box 112612, United Arab Emirates; 4Riphah Institute of Pharmaceutical Sciences, Ripha International University, Islamabad 44000, Pakistan; 5Department of Pharmaceutics, College of Pharmacy, King Saud University, P.O. Box 2457, Riyadh 11451, Saudi Arabia; 6School of Veterinary Medicine, University College Dublin, Belfield, D04 C1P1 Dublin, Ireland; 7Department of Pharmacy, COMSATS University Islamabad, Lahore Campus, Defense Road, 1.5 Km Off Raiwind Road, Lahore 54000, Pakistan; 8College of Pharmacy, University of Sargodha, Sargodha 40100, Pakistan; mulazim.hussain@uos.edu.pk

**Keywords:** thiomers, cyclodextrin, hair keratin, minoxidil, hair adhesion

## Abstract

Purpose: The aim of this research was to prepare adhesive nanoparticles for the topical application of Minoxidil (MXD). Methods: Thiolated β-CDs were prepared via conjugation of cysteamine with oxidized CDs. MXD was encapsulated within thiolated and unmodified β-CDs. Ionic gelation method was used to prepare nanoparticles (Thio-NP and blank NP) of CDs with chitosan. Nanoparticles were analyzed for size and zetapotential. Inclusion complexes were characterized via FTIR. Drug dissolution studies were carried out. An in vitro adhesion study over human hair was performed. An in vivo skin irritation study was performed. Ex vivo drug uptake was evaluated by using a Franz diffusion cell. Results: Thiolated β-CDs presented 1804.68 ± 25 μmol/g thiol groups and 902.34 ± 25 μmol/g disulfide bonds. Nanoparticles displayed particle sizes within a range of 231 ± 07 nm to 354 ± 13 nm. The zeta potential was in the range of −8.1 ± 02 mV, +16.0 ± 05 mV. FTIR analyses confirmed no interaction between the excipients and drug. Delayed drug release was observed from Thio-NP. Thio-NP retained over hair surfaces for a significantly longer time. Similarly, drug retention was significantly improved. Thio-NP displayed no irritation over rabbit skin. Conclusion: Owing to the above results, nanoparticles developed with MXD-loaded thiolated β-CDs might be a potential drug delivery system for topical scalp diseases.

## 1. Introduction

Hair loss is an emblematic human scalp-related condition caused by chemotherapy, chronic diseases and aging. Androgenetic alopecia (AGA) is a genetic disorder of hair loss that has been observed in both sexes [1]. At present, it is estimated that approximately 50% of males and 15–30% of females are facing hair loss problems [2]. Minoxidil (MXD) is considered a drug of choice for the topical treatment of hair fall in men and women caused by a disorder known as AGA. MXD is a derivative of pyrimidine that acts as a peripheral vasodilating agent, and initially in 1970, it was approved for reducing blood pressure [3,4,5]. MXD opens the potassium channel, releases nitric oxide (NO) and increases the flow of blood towards the follicles of hair, modifying the pathway of prostaglandins and consequently suppressing the pathway of alopecia [4,5]. Chronic oral intake of MXD could cause adverse skin reactions, which cause the discontinuity of therapy and decrease patient adherence to therapy. Similarly, as the water solubility of MXD is very low, problem of crystallization arises upon evaporation of the solvents, when MXD is solubilized using nonpolar solvents like ethanol and propylene glycol, which may persuade several adversative effects, such as pruritus, rash, redness of skin, dandruff, dryness of scalp and allergic contact dermatitis [6]. Therefore, distinctive, and suitable new formulations are required for the topical application of MXD [5,7].

To address the issues argued above, previously developed formulations for topical MXD application include 5% minoxidil solution, nanoparticles with penetration enhancers and mucoadhesive polymer-based formulations [5,8]. Plenty of active medicine is lost in the course of the application of MXD solutions, and poor adherence towards therapy is associated with this, which could further lead to ineffective dose regulation [5,9]. Likewise, a penetration enhancer could lead to the accumulation of toxic levels of MXD in the skin tissue, and mucoadhesive formulations cause stickiness over the scalp and area of application [5].

Cyclodextrins (CDs), cyclic polysaccharides with 6–8 units of glucopyranose, are enzymatically produced by the mortification of starch [5,10]. CDs have a lipophilic cavity for the encapsulation of active drug molecules, and CDs encourage the sustained and extended release of encapsulated model drugs. CDs have been recognized as the smallest known drug carriers and are capable of enhancing the dissolvability of numerous drug molecules by the development of comparatively persistent inclusion complexes. CDs have a high affinity for a particular drug molecule because of their carrier system, which equips them with a specific drug release profile [11,12]. Previous studies from our research group have demonstrated that thiolated cyclodextrins provide the advantage of covalent bond formation through thiol/disulfide interchange reactions with cysteine-rich subregions on hair surfaces or other dermal appendages. Therefore, the aim of this research was to develop inclusion complexes of MXD with thiolated β-CD-SH [13]. Thiol moieties attached to β-CD-SH develop covalent bonds with the disulfide bonds of alpha keratin. The inclusion complex of MXD with β-cyclodextrin will be formed to enhance its water solubility, which will further improve its activity.

## 2. Materials and Methods

### 2.1. Materials

Cyclomaltoheptaose known as Beta-cyclodextrin (CID-444041), Cysteine HCl (CID-60960), Ellman’s reagent (3-Carboxy-4-nitro phenyl disulfide) (CID-6254), MES hydrate (Sodium 2-morpholino ethane sulfonate) (CID-23673676), cysteamine (CID-6058), sodium periodate (CID-23667635), sodium cyanoborohydride (CID-20587905), dimethyl sulfoxide (DMSO), ethylene glycol, minoxidil (MXD), distilled water, chitosan, and sodium phosphate monobasic dihydrate (purity ≥ 99%) were obtained from Glentham Life Sciences, UK. Analytical grade chemicals and reagents were purchased from International chemical suppliers. Pretreated regenerated cellulose dialysis tubing of 1000–2000 Da was purchased from Spectrum, USA.

### 2.2. Method

#### 2.2.1. Oxidation

To develop aldehyde functional groups in the structure of β-CD, an oxidation reaction was carried out. Briefly, a 1% solution of β-CD was prepared in 180 mL deionized water in aluminum-wrapped 500 mL Erlenmeyer flasks. Sodium per iodate (NaIO_4_, 0.8 g) dissolved in 20 mL deionized water was added to the above solution to create di-aldehyde functional groups following the process of oxidation. The solution mixture was vigorously mixed by stirring at 25 °C for almost 2 h, to consume the unreacted NaIO_4,_ 800 µL of ethylene glycol was added. Finally, the reaction mixture was stirred at 25 °C for 1 h, and then the final product (Aldo-β-CD) was isolated via dialysis against distilled water for 72 h. During the dialysis process, the dialyzing water was replaced three times a day. The purified product was freeze-dried to produce the final product [13,14,15].

#### 2.2.2. Conjugation of Aldo-β-CD with Cysteamine HCl

For the conjugation reaction, 1% Aldo-β-CD was dissolved in 0.1 M buffered solution of 4-morpholine-ethanesulfonic acid (MES) hydrate. After equilibration, 0.5% cysteamine HCl dissolved in deionized water was added. 0.01 M HCl was used to bring the pH of the mixture to 4. The ultimate volume of the mixture was maintained at 100 mL with purified water. The reaction mixtures were stirred in the dark at 25 °C for 3 h. NaCNBH_3_ (8%) was added, and the reaction was stirred for an additional 72 h beneath a laminar air flow hood. A similar method was used for the conjugation of the fluorescent marker rhodamine-123. Once the reaction was finished, the final solutions were dialyzed almost 6 times at 10 °C by using a dialyzing tube to separate the polymer conjugates and unreacted residues of chemicals. In detail, the mixture was dialyzed 2 times at pH 4.5 against distilled water and then 2 times against distilled water at pH 4.5, but 1% sodium chloride was added to quench the ionic interactions. Finally, samples were dialyzed 2 times against distilled water at pH 4. Finally, the purified solutions obtained after dialysis were frozen at −78 °C and lyophilized under vacuum for 1 day. Dried conjugates were stored at 4 °C in hermetic containers [13].

### 2.3. Encapsulation of MXD

Encapsulation of MXD within thiolated and unmodified CDs was carried out by solvent evaporation method, briefly, 25 mg of MXD dissolved in 5 mL ethanol was dispersed in 30 mL of phosphate sodium buffer (0.05 M) containing 75 mg thiolated β-CD (Thio-β-CD) and unmodified β-CD in different molar ratios with increasing concentrations of MXD, ranging from 0.5:1 to 4:1, and the mixtures were continuously agitated at 25 °C for 24 h. Finally, the mixture suspensions were strained through a porous cellulose membrane filters with a pore size of 0.45 µm, and the filtrate obtained was freeze and lyophilized [13,16]. MXD entrapment was determined by measuring the amount of MXD in 15 mg of inclusion complexes.

### 2.4. Preparation of Nanoparticles

The nanoparticles were prepared by using a previously reported method of ionic gelation [17]. Briefly, Chitosan (15 mg/mL) dissolved in acetic acid (1% *v*/*v*) was continuously stirred with 1 mL (10 mg/mL water) solution of inclusion complex of Thio-β-CD (MXD loaded CD). Five Thio-NP formulations as shown in Table 1, were prepared by varying the amount of MXD loaded Thio-β-CD. Encapsulated solution with increasing concentrations of MXD loaded Thio-β-CD was added dropwise into the chitosan solution and continuously stirred. The pH of all mixtures adjusted to 4 and continuously stirred at 37 °C for 60 min, then sodium tri-polyphosphate TPP (3 mL, 4 mg/mL) was added dropwise. Similarly, blank NP were prepared with unmodified β-CD loaded with MXD. Likewise fluorescently labeled Thio-NP and blank NP were prepared by adding fluorescently labeled β-CD. The obtained nanomaterials were filtered to separate both insoluble and soluble material. Then, the water phase was neutralized, and dialysis was performed for 24 h. Yellowish colored product was yielded after drying the solution under reduced pressure. Nanoparticles developed by using Thio-β-CD were labeled Thio-NP, and nanoparticles formulated with unmodified CD were labeled blank-NP.

### 2.5. Thiol Group Determination and Estimation of Disulfide Content

For the quantification of thiol groups conjugated on Thio-β-CD, Ellman’s reagent was used; likewise, the amount of thiol groups present over the surface of nanoparticles was also determined [18]. Similarly, disulfide contents were also determined after reduction with sodium borohydride (NaBH_4)_ and the addition of 5,5′-dithiobis (2-nitrobenzoic acid) [19].

### 2.6. Surface Properties and Physical Characterization of Inclusion Complexes and Nanoparticles

Thiolated and unmodified β-CD were characterized by ^1^H-NMR using Gemini Varion 200 NMR spectrometer. FT-IR analysis was used for the confirmation of inclusion complex formation between MXD and CDs. Encapsulation of MXD was confirmed by comparing the variance of peak dimensions, position, and strength. FTIR spectroscopy is also used to confirm the possible physical and chemical interactions among β-CD and MXD in the solid form because complex formation will change the absorption spectrum of MXD [12]. FT-IR spectra of powdered Thio-β-CD, MXD encapsulated within Thio-β-CD and free MXD were measured by benchtop FTIR spectroscopy.

### 2.7. Scanning Electron Microscopy for Surface Properties of Nanoparticles

The surface properties of nanoparticles prepared from thiolated CDs (Thio-NP) and with unmodified Blank NP were evaluated by SEM, (Zeiss, model (DSM-940A), Oberkochen, Germany) operating between 5 and 20 kV. The sample of lyophilized nanoparticles was mounted on metal stubs in nitrogen environment, gold was coated over particles. Finally, the scans were acquired and projected by appropriate software.

### 2.8. Zeta-Potential Measurements

The particle size of all six formulated nanoparticles (Thio-NP, Blank-NP) was measured by using Malvern, Panalytical Zeta Sizer, UK. The size measurement was optimized by software equipped with ZS Xplorer software. For the evaluation of particle size, 50 µg of Thio-NP and Blank-NP formulations were suspended in deionized water. Each sample was run for 30 sec each for ten cycles of measurement. The measurement process for particle size evaluation was repeated three times. All the samples were analyzed at 25 °C, and other parameters, such as the electric field at 13.89 V/cm, refractive index at 1.330, and voltage at 5 V. The mean zeta potential with SD was computed utilizing the ZS Xplorer software. All results are the mean ± SD of three different replicates and measurement were repeated 10 time [20].

### 2.9. Dissolution Study

For this purpose, 30 mg of nanoparticles formulations (Thio-NP-1 to Thio-NP-5) containing MXD-loaded thiolated β-CD and blank-NP containing MXD equivalent to 20 mg of pure MXD were weighed and compressed into tablets. Freshly prepared 0.1 M phosphate buffer (pH 6.8) was employed as a dissolution medium. The samples were transferred to 10 mL of phosphate buffer in Erlenmeyer flasks. The Erlenmeyer flasks with NP were placed in an incubator on a rocker at 37 °C and at comparative humidity maintained at 100%. The sink condition was retained during the test study. Specimens of 200 µL were withdrawn at scheduled time intervals, and the volume was replaced with an equal volume of fresh phosphate buffer. Centrifugation and filtration of samples was carried out before analysis. All the samples were analyzed using HPLC equipped with a ProntoSIL C-18 column. The quantification of MXD was performed by comprehending peaks, and the concentration of the soluble drug was calculated by approximation from a model curve made of higher strengths of MXD. Collective rectification was made for already drawn samples [13].

### 2.10. In Vitro Evaluation of Adhesion Properties on Human Hairs

For the adhesive properties, Thio-NP prepared with fluorescently labeled thiolated β-CD, human hair cuttings were collected from a barbershop located in Shershah block A. As the hair cuttings at barbershops in Pakistan are generally considered as a waste, therefore ethical committee waived the condition of taking any written or verbal consent. For preparing the hair cutting for adhesive studies, a phosphate buffer (100 mM) with pH 6.8 was employed to wash and rinse the hair cuttings continuously. Approximately 30 mg of fluorescently labeled Thio-NP and Blank-NP were equally applied all over the length of hair after a stabilization time of 5 min. Hairs were placed at 45°-degree angle, continuously rinsed with phosphate buffer by a peristaltic pump with a persistent surge rate of 1 mL/min was used as show in Figure 1. After a stabilization time of 5 min, sample collection was initiated at fixed intervals of 30, 60, 90, 120, and 180 min, and the phosphate buffer that flowed down the hair was gathered. The similar process was repeated on the hair for 0.2 mg of free rhodamine-123, but without any polymer, and later, to equate with Thio-NP and Blank-NP, 30 mg of thiolated and unmodified β-CD (drug free) was added to the phosphate buffer that had been collected and considered a standard for the measurement of adhesiveness. All the collected specimens were vortexed for 5 min, after which 100 µL from every sample was transmitted to a microplate reader, and the absorption was documented at a wavelength of 535 nm and excitation at a wavelength of 485 nm. All experiments will be performed in triplicate [13].

### 2.11. Skin Tolerance Test

To perform in vivo skin corrosion and irritation tests, healthy rabbits with intact skin were utilized (age = 3.9 months, average weight = 3.0 kg). Ethical approval for conduction of animal study was obtained from COMSATS research ethics and biosafety committee via letter number CUI-REB/23. Twenty-four hours prior to performing the test, an electrical shaver was used to remove hair from 3 areas of an estimated 6 cm^2^. A pre-weighed quantity of 500 mg of all six nanoparticle formulations (Thio-NP-1 to Thio-NP-1-5, Blank-NP) were uniformly dispersed on a lint fabric, applied carefully to the shaved area of rabbit, and shielded with a gauze patch, which was kept in position by using a nonirritant flexible patch. If no skin reaction was noticed in the first 180 s after applying the NP loaded cloth, the second patch was administered at another shaved area and removed after 60 min. If the findings at this point suggest that exposure to nanoparticles for up to 60 min was quite normally bearable without any irritation, a third patch was added and removed after 4 h. The reaction was calculated in accordance with the guidelines [21]. At the end of the experiments, nanoparticles were removed from the skin, all the rabbits were active and healthy, moreover, rabbits were kept isolated in the departmental animal house for further 14 days for further observation of any kind of skin allergy.

### 2.12. Drug Uptake through Skin

Drug uptake across the bovine ear skin membrane was evaluated by using Franz diffusion cells by following the methods reported in earlier studies, and the thickness of the bovine skin used for permeation ranged from 4 to 6 mm. Bovine ear skin was acquired from Khaliq slaughterhouse, situated at main road, Sher shah Raiwind Road, Lahore. Before the experiment, skin was prepared by removing fats and extra subcutaneous layers by using a scalpel. The final isolated and prepared skin membrane was stored at −20 °C. Briefly, the receptor compartment of the Franz diffusion cell was filled with 8.5 mL of phosphate buffer (pH 6.8) containing 1% Tween 80 to maintain sink conditions. The prepared skin membrane was clamped between the upper and lower compartments of the Franz cell. Fifty milligrams of optimized nanoparticle formulations (Thio-NP-2 and Blank-NP) containing an equivalent quantity of 20 mg MXD and, similarly, ethanolic solution containing 20 mg MXD (Control) were added to the upper donor chamber.

Uptake and drug retention experiments were carried out for 180 min. Samples from the receptor compartment were withdrawn at regular intervals of 30 min. Equal volumes of buffer were replaced with each sample withdrawal. At the end of the 3 h experiment, formulations from the donor chamber were removed, and the skin membrane was cleaned with isopropyl alcohol. To evaluate and measure the amount of MXD retained within skin and follicles, differential stripping, and coating of skin with acrylate. During stripping of skin, initial stripping was not considered for analysis. All the tape strips, acrylate coating and skin tissue were dipped in ethanol for extraction of MXD, and all the suspensions and skin tissues were centrifuged for 30 min at 14,000 rpm. The concentration of MXD at different intervals and extracted from strips and acrylate coatings was analyzed by HPLC.

### 2.13. Statistical Data Analysis

Statistical data analysis was executed by utilizing Student’s t test with a confidence interval (*p* < 0.05) for analyzing two test groups.

## 3. Results and Discussion

### 3.1. Characterization of β-CD-Cys Derivatives

To induce the mucoadhesive properties to β-CD oligomers, thiolation was carried out via the oxidation and subsequent reductive deamination of β-CD. Thiolation of β-CD was carried out successfully by the covalent binding of cysteamine to β-CD-CHO. Reaction of functional groups such as primary amines present in cysteamine requires the presence of carbonyl groups on β-CD, however unmodified β-CD does not have such carbonyl groups needed for the reaction with primary amines. For the oxidation of β-CD, NaIO_4_ was used, and the conjugation of cysteamine with β-CD-CHO was achieved by a reducing agent, known as NaCNBH_3_. The process of thiolation of CD via oxidation method and via substitution reaction method are well established methods previously validated via ^1^H-NMR as shown in (Figure 1a,b) and chemical analysis techniques, by various researchers including research work from our group [12,13,22]. Increasing the concentration of NaIO_4_ could lead to an increase in the oxidation state of β-CD and subsequently increase the number of conjugated cysteamines [13]. Thiolated β-CD, obtained after lyophilization, was an amorphous, odorless, and white powder. Inclusion complexes with thiolated β-CD showed better drug encapsulation properties due to disulfide bridges among the individual CDs oligomers, 68% loading and 47% entrapment efficiency was found with thiolated β-CD. Owing to encapsulation of lipophilic drugs CDs have outstanding applications in pharmaceutical industry, and now β-CD oligomers are most commonly been employed in topical delivery systems [23].

### 3.2. Determination of Thiol Group and Disulfide Bond Contents

The thiolated conjugates obtained were characterized via Ellman’s reagent, and the results showed 1804.68 ± 25 μmol/g free thiol groups and 902.34 ± 25 μmol/g disulfide bonds, on average, as shown in Table 2. Likewise, increasing the concentration of cysteamine while keeping the NaIO_4_ concentration constant could lead to an increased number of thiol groups attached, as is evident from the research work of previous researchers [13].

### 3.3. FTIR Spectral Analysis of Thio-β-CD

FTIR analysis of the cysteamine conjugation and confirmation of inclusion complexes was carried out. The deviation in the absorption peaks of IR, like their shape, changes in shift, and intensity absorption peaks of the guest or host can grant sufficient evidence for the occurrence of the inclusion complex. The presence of thiol groups was established by the results acquired from FTIR spectroscopy. The FTIR spectra of MXD complexed with thio-β-CD and unmodified β-CD are shown in Figure 2. A small and floppy vibrational stretch witnessed at approximately 2550–2625 cm^−1^ was allocated to the -SH stretching vibration in thio-β-CD. The expansive absorption band at approximately 3100–3550 cm^−1^ was credited to -NH (which also confirms the presence of conjugated cysteamine) and -OH groups co-existing within thio-β-CD [13].

In the case of aldehydic β-CD, the existence of -OH groups could be confirmed by the absorption displayed at 3150–3400 cm^−1^, which is characteristic of -OH groups presence. The C-H bonds present in CDs could be confirmed by the absorption band at 2900 cm^−1^. Similarly, the absorption at 1680 cm^−1^ was recognized to aldehyde groups of the studied compounds. The absorption band at 1654 cm^−1^ vanished or shifted to low wavenumbers in the MXD-β-CD inclusion complex, indicating that the C−O stretching vibration was confined after the formation of the inclusion complex. 1552 cm^−1^ was greatly weakened, indicating that a majority of MXD was included in the complex. The FTIR spectra of free minoxidil and β-CD are presented in Figure 3 for comparative analysis.

### 3.4. Characterization of Surface Properties and Potential Charges of NPs

All the nanoparticles prepared from thiolated (Thio-NP-1 to Thio-NP-5) and unmodified β-CD (Blank-NP) displayed uniform particle sizes within the range of 231 ± 07 nm to 354 ± 13 nm. In general, the Z-average diameters for Thio-NP composed of thiolated β-CD were larger than the core size of the nanoparticle (Blank-NP) consisting of unmodified β-CD because of the disulfide bond formation and agglomeration of thiolated β-CD; on the other hand, the nanoparticles composed of unmodified β-CD have a smaller size. Increased size with the increasing concentration MXD could be the reflection of increased amount of drug loading within the complexes of the thiolated β-CD. Similarly, the zeta potential of all the nanoparticles was in the range of −8.1 ± 02 mV, +16.0 ± 05 mV and the significant shift of negative potential to positive potential seems due to the cat-ion charges of –NH^+^ present in the amino groups of cysteamine attached to thiolated β-CD [24]. SEM projections produced from electron microscopy of Thio-NPs (shown in (Figure 1c,d) predict spherical and semicircular structures indicating smooth and stable NPs due to possibility of cationic surface charges.

### 3.5. Drug Dissolution Studies

The dissolution profiles of Thio-NP and blank-NP formulations were evaluated and compared with the calibration curve of standard MXD. The drug release profile of Thio-NP tested conceivably validates the impact of the realistic effect of thiolation and the potential for multi purposing of Thio-NP composed of thiolated β-CD and chitosan. The calibration curve shown in Figure 4 was developed with increasing concentrations of MXD. Notably, this is one of the few studies that has evaluated the topical drug release of nanoparticle formulations. Therefore, the oral formulations and the drug properties control the absorption after oral administration, as evidenced by the incongruity between the plasma concentration of one of the drugs and the in vitro release kinetics [25]. In contrast to the available research, it has been revealed here that excipient-free β-CD nanoparticles can be cast off as release-retarding excipients and to prepare nanocarriers with desired release profiles. Nanoparticles prepared with high concentration of MXD loaded thiolated β-CD (Thio-NP-4, Thio-NP-5) displayed initial quick drug release followed by slow release of MXD, as shown in Figure 5, which could be linked with excessive availability of encapsulated drug to water penetration and fast dissolution. Moreover, increasing the inclusion complex contents in NP formulation led to increase in the size of particles, which could be a reason of initial quick drug release. In the case of Thio-NP-3, we observed sustained drug release for the first 1–3 h, because drug entrapment within the CD adjoining the oligomers prohibited water from penetrating the internal core containing the drug. After the first phase of release due to dissolution of the nanoparticles, the drug-loaded β-CD functioned as a 2-h long stop-gate for the second phase of nanoparticles to dissolve and release the drug [26].

### 3.6. In Vitro Evaluation of Adhesion Properties on Human Hairs

Topically applied nanoparticles in the form of aqueous suspensions are pharmaceutical persistent release formulations that maneuver in a state firmly attached to human hairs; consequently, adhesion properties are imperative characteristics of this form of application. There are many concepts of adhesion and fundamental forces that appear to act independently [27]. In previous adhesion studies over porcine skin/hairs, skin adhesion was measured from the pure unmodified β-CD combinations of straight compressed discs as well as from the final drug composition. The outermost coating of the skin is the stratum corneum, which consists of both lipophilic and hydrophilic domains as well as hair follicles and sweat glands. The hydrophilic parts consist mostly of keratin. The circumstance that high polarity formulations such as hydrogels may also adhere to skin may be in part due to the skin being wet and perhaps due to the presence of hair follicles and sweat glands, which contain aqueous channels [28]. However, in the case of thiolated CDs containing nanoparticles (Thio-NP), disulfide bonds are formed between the hair keratin and free thiol groups attached to the CDs [29]. These free thiol groups develop strong covalent bonds with the hair roots, and in this way, nanoparticles reside over hairs for a longer time, as shown in Figure 6, in which nanoparticles formulated with thiolated CDs were retained for a significantly longer time than nanoparticles consisting of unmodified CDs.

### 3.7. Skin Tolerance Test

A skin tolerance test was executed on rabbits to appraise whether the application of newly developed adhesive nanoparticle suspensions might cause skin irritation [30]. All nanoparticle formulations developed from thiolated (Thio-NP) and unmodified CDs (Blank-NP) showed no significant skin irritations (erythema and edema) on normal rabbit skins. Figure 7 shows photographs of sections of the hairless skin of rabbits treated with unmodified CDs nanoparticles (blank-NP), skin treated with thiolated CDs nanoparticles (Thio-NP-3) and skin treated with the strong base NaOH. It was evident that skin samples treated with nanoparticles developed from thiolated CDs did not reveal any marked changes in the normal skin surfaces and no signs of inflammation, guaranteeing good skin tolerability of the developed formulation [31].

### 3.8. MXD Uptake through Skin

The amount of drug accumulated in the skin compartments, expressed as a percentage of the dose applied on the skin surface, is shown in Table 3. Through this experiment, the main objective was to evaluate whether nanoparticles with thiolated CDs (Thio-NP-3) would remain attached for a longer duration at the hair shaft compared with nanoparticles developed with unmodified CDs (Blank-NP). In this context, the Thio-NP demonstrated greater MXD retention in dermal layer compared with the greater penetration of blank NP in the hair follicles, and the least drug penetration was observed for the Thio-NP sample (p < 0.05), which reflects the attachment of thiol groups with keratin (highly rich in disulfide bonds) present in the hair shaft and therefore, increased the retention of MXD at the application side, which led to an increased accumulation of MXD loaded within thiolated nanoparticles [32,33]. The effect of thiol groups present in Thio-NP on hair keratin targeting and the subsequently increased retention effect is already well known [34,35,36]. Moreover, it is noteworthy that the presence of sebum in the follicle compartment can particularly increase the partition and permeation of (MXD) hydrophobic compounds [37].

Thio-NP and blank-NP provided significantly different transdermal drug permeation rates, with 7.35 ± 1.67% and 3.56 ± 0.34% of the total doses of MXD applied on the skin surface, respectively, with significant differences among these results (*p* > 0.05). The significantly improved MXD dermal layer retention observed for Thio-NP is reflected in MXD dermal accumulation and lesser follicular penetration. Regarding follicular penetration, Thio-NP provided 1.56 ± 0.57% of the total amount of MXD, whereas Blank-NP provided 8.45 ± 2.69%. These results show that Thio-NP could retain and maintain higher concentrations of MXD at the hair shaft for at least 3 h, which might provide higher efficacy in the treatment of AGA.

## 4. Conclusions

This research work has revealed that nanoparticles developed from thiolated CDs can be premeditated to adhere over hair shafts for a prolonged time. CDs anchored with thiol groups when formulated into nanoparticles can bind to sulfur moieties of keratin and are confined on the skin external and near to the skin exterior in the hair follicle. For nanoparticles ornamented with thiolated CDs, roughly half of the thiol groups appeared over the particle surface, so binding to keratin was greatly increased. Nanoparticles developed from thiolated CDs displayed uniform particle size and positive zeta potential, and MXD loaded in the CDs showed sustained release. By controlling the amount of the free thiol groups attached over the CDs, it is reasonable to develop nanoparticle formulations focused over the hair shafts and might be a potential carrier for application for male baldness issues and other skin topical issues.

## Data Availability

Data are contained within the article.

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
