# Peer review of "Development and Characterization of Thiolated Cyclodextrin-Based Nanoparticles for Topical Delivery of Minoxidil"

_pharmaceutics, 2023, doi:10.3390/pharmaceutics15122716_

Round 1
Reviewer 1 Report
Comments and Suggestions for Authors
The authors have developed and characterized thiolated cyclodextrin as topical formulation for delivery of minoxidil. The Preparation process wis described in detailed and thiolation is characterized by FTR spectroscopy. The invitro adhesion study and invitro skin irritation study performed depicts good binding ability to the hairs and good skin tolerability. The overall structure of the manuscript is good and could be accepted for publication with the following minor comments.
1. Initially incubation was started with thiolated β-CD (Thio-β-CD) and unmodified β-CD in a molar ratio of 1:1. Di the authors tried any other molar ratio for better encapsulation efficiency.
2. Authors could pictorially represent the invitro adhesion properties on human Hairs. How it was carried out as the study looks very interesting and provides a method to study invitro adhesion of topical formulations prepared for hair loss.
3. Authors have mentioned the particle size 231 nm to 319 nm or zeta potential ̶8.1 to + 12.0, instead they could consider mentioning the particle size as Mean±SD. Also, the units for zeta potential could be added.
4. Results and discussion section the authors could cite the latest work for topical formulations for scalp diseases using cyclodextrin.
5. The developed thiolated cyclodextrin will be further developed into any formulations like cream, shampoo or ointments better applicability and patient compliance.

Comments on the Quality of English LanguageAuthor Response
Saraniya Ramachenderan
Editor
Pharmaceutics, MDPI
Response Letter to the Pharmaceutics-2679697
Dear Editor,
Thank you very much for sending us the reviewer’s comments on our manuscript entitled “Development and Characterization of Thiolated Cyclodextrin-Based Nanoparticles for Topical Delivery of Minoxidil”.
Additionally, we would also like to thank the reviewers for their encouraging feedback and the decision to give us an opportunity for a revision according to their constructive discussion of weaknesses in our manuscript.
This response letter summarizes our revision and addresses the points raised by the reviewers and editor.
Comments are printed in bold followed by their answers while changes in the revised manuscript are highlighted in blue.
Reviewer 01
The authors have developed and characterized thiolated cyclodextrin as topical formulation for delivery of minoxidil. The Preparation process wis described in detailed and thiolation is characterized by FTR spectroscopy. The invitro adhesion study and invitro skin irritation study performed depicts good binding ability to the hairs and good skin tolerability. The overall structure of the manuscript is good and could be accepted for publication with the following minor comments.
- Initially incubation was started with thiolated β-CD (Thio-β-CD) and unmodified β-CD in a molar ratio of 1:1. Di the authors tried any other molar ratio for better encapsulation efficiency.
Ans: According to the suggestions of reviewers, authors have tried different molar ratios of MXD with CD to assess the effect of different molar ration on encapsulation efficacy, but it was observed that molar ratio 1:1 yielded the maximum encapsulation. Moreover, our group had published lots of CD encapsulation related research work and most of the studies demonstrate that CDs, owing to one lipophilic cavity, showed maximum drug loading at 1:1 ratio relevant reference have cited within article. .
- Authors could pictorially represent the invitro adhesion properties on human Hairs. How it was carried out as the study looks very interesting and provides a method to study invitro adhesion of topical formulations prepared for hair loss.
Ans: According to the suggestion pictorial illustration have been added.
- Authors have mentioned the particle size 231 nm to 319 nm or zeta potential ̶8.1 to + 12.0, instead they could consider mentioning the particle size as Mean±SD. Also, the units for zeta potential could be added.
Ans : According to suggestions, means SD have been added and units for zeta potential are added. 231±07, 319±13, -8.1±02 mV, +12.0±05 mV
- Results and discussion section the authors could cite the latest work for topical formulations for scalp diseases using cyclodextrin.
Ans : According to suggestions latest work have been cited.
- The developed thiolated cyclodextrin will be further developed into any formulations like cream, shampoo or ointments better applicability and patient compliance.
Ans: We are planning to develop some topical sprays, or shampoos for commercialization purposes.
Reviewer 2 Report
Comments and Suggestions for Authors
The reviewed manuscript presents the results of research on the formation of Minoxidil nanocarriers made from cyclodextrin derivatives for use in anti-alopecia therapy. The work is written correctly and proves the authors' proficiency in researching the course of drug release from polymer matrices. Unfortunately, the scope of research presented in the manuscript is too limited for this work to be published in this form in Pharmaceutics. It is necessary to expand it to include;
- more detailed characterization of the properties of the obtained cyclodextrin derivatives, including individual stages of synthesis illustrated by changes in NMR spectra and the appearance or change in the position of characteristic signals. In this case, the presented FTIR spectra do not contribute much. The actual chemical structure of the obtained compounds must be convincingly proven using multiple measurement methods;
- the characterization of nanoparticles is incomplete, especially regarding issues such as; MXD encapsulation efficiency, the influence of MXD content on the characteristics of the obtained particles;
- supplementing the research with attempts to measure the course of drug release from the obtained nanoparticles with different MXD content, optimization of the release system, ensuring the proper course of the planned therapy;
Without developing the above issues, the work is too poor and unsuitable for publication due to its low scientific value.
Comments on the Quality of English LanguageThe text is readable, written clearly, without major grammatical, syntactical or grammatical errors.
Author Response
Saraniya Ramachenderan
Editor
Pharmaceutics, MDPI
Response Letter to the Pharmaceutics-2679697
Dear Editor,
Thank you very much for sending us the reviewer’s comments on our manuscript entitled “Development and Characterization of Thiolated Cyclodextrin-Based Nanoparticles for Topical Delivery of Minoxidil”.
Additionally, we would also like to thank the reviewers for their encouraging feedback and the decision to give us an opportunity for a revision according to their constructive discussion of weaknesses in our manuscript.
This response letter summarizes our revision and addresses the points raised by the reviewers and editor.
Comments are printed in bold followed by their answers while changes in the revised manuscript are highlighted in blue.
Reviewer 02
The reviewed manuscript presents the results of research on the formation of Minoxidil nanocarriers made from cyclodextrin derivatives for use in anti-alopecia therapy. The work is written correctly and proves the authors' proficiency in researching the course of drug release from polymer matrices. Unfortunately, the scope of research presented in the manuscript is too limited for this work to be published in this form in Pharmaceutics. It is necessary to expand it to include.
- more detailed characterization of the properties of the obtained cyclodextrin derivatives, including individual stages of synthesis illustrated by changes in NMR spectra and the appearance or change in the position of characteristic signals. In this case, the presented FTIR spectra do not contribute much. The actual chemical structure of the obtained compounds must be convincingly proven using multiple measurement methods.
Ans: Thank you very much for the suggestions of more detailed characterizations. Our research group has published various Thiolated CDs related to research work and have performed characterization of different synthetic stages via NMR. So, authors considered it is not necessary to repeat the previously analytical test to validate already validated structures and methods of thiolation. Our focus in this research work is to develop a nanoparticle based topical delivery system for minoxidil, that’s why our analytical focus and characterization remained focused mostly for NP evaluations. As a ref relevant article have been cited here.
- the characterization of nanoparticles is incomplete, especially regarding issues such as MXD encapsulation efficiency, the influence of MXD content on the characteristics of the obtained particles; - supplementing the research with attempts to measure the course of drug release from the obtained nanoparticles with different MXD content, optimization of the release system, ensuring the proper course of the planned therapy.
Ans: According to suggestions, 5 nanoparticles formulations with increasing concentration of MXD loaded CD were developed and evaluated for NP size and zeta potential changes. Moreover, dissolution study was also repeated with new formulations and results have been added in article.
Without developing the above issues, the work is too poor and unsuitable for publication due to its low scientific value.
The text is readable, written clearly, without major grammatical, syntactical, or grammatical errors.
Round 2
Reviewer 2 Report
Comments and Suggestions for Authors
My previous comments were taken practically into account by the authors. Only the conclusion part lacks a general conclusion regarding the answer to the question, which of the obtained drug nanocarriers seems optimal for the application in inquiry and why? I would ask the authors to introduce this summary at the end of this work.
Comments on the Quality of English LanguageI didn't notice any syntax or grammatical errors.
Author Response
We are again thankful for suggestion about the conclusive summary.
The follwing lines highlighted in maron color have been added to the conclusion part of manuscript.
Adhesive nanoparticles, Thio-NP-3 developed from thiolated CDs displayed uniform particle size and positive zeta potential, furthermore, MXD loaded in the optimized Thio-NP-3 showed sustained release. By controlling the amount of the free thiol groups attached over the CDs, it is reasonable to develop optimized nanoparticle formulations with sustained drug release and dedicated to significantly improved topical adhesion over the hair shafts. Based on above findings, it could be summarized that nanocarriers based on thiolated CDs might be potential carriers for application for male baldness issues and other skin topical issues.